# Deletion Syndrome 22q11.2: A Systematic Review

**DOI:** 10.3390/children9081168

**Published:** 2022-08-03

**Authors:** Jonathan Cortés-Martín, Nuria López Peñuela, Juan Carlos Sánchez-García, Maria Montiel-Troya, Lourdes Díaz-Rodríguez, Raquel Rodríguez-Blanque

**Affiliations:** 1Research Group CTS1068, Andalusia Research Plan, Junta de Andalucía, 18014 Granada, Spain; jonathan.cortes.martin@gmail.com (J.C.-M.); mariamontiel@ugr.es (M.M.-T.); cldiaz@ugr.es (L.D.-R.); raquel.rodriguez.blanque.sspa@juntadeandalucia.es (R.R.-B.); 2Department of Nursing, School of Health Sciences, University of Granada, 18016 Granada, Spain; 3Vall D’hebron University Hospital, 08035 Barcelona, Spain; nurialp268@gmail.com; 4Department of Nursing, School of Health Sciences, Ceuta Campus, University of Granada, 51001 Ceuta, Spain; 5San Cecilio University Hospital, 18016 Granada, Spain

**Keywords:** 22q11.2 deletion syndrome, DiGeorge syndrome, velocardiofacial syndrome, rare disease, congenital anomalies, daily activities

## Abstract

22q11.2 deletion syndrome (DS 22q11.2) is a rare disease of genetic origin, caused by the loss of the q11.2 region of chromosome 22. It affects one in 4000 live newborns, and among the clinical manifestations that can occur in this syndrome are abnormalities in the parathyroid glands (producing calcium deficits), the palate, the heart and the thymus. It is also known as DiGeorge syndrome or velocardiofacial syndrome, among other names, depending on the clinical presentation of each individual. The main objective of the review was to update information on DS 22q11.2 from publications in the scientific literature. The daily activities of these patients are seriously impaired, due to the impact of the clinical manifestations. Interventions can be performed to improve their social, cognitive and emotional skills, thus increasing their ability to perform different daily activities.

## 1. Introduction

Deletion syndrome 22q11.2 is a rare disease [1] caused by a deletion on chromosome 22, specifically at locus q11.2. In addition to this syndrome, there are three other types of abnormalities that can affect this area of the chromosome in different ways: terminal microdeletion syndrome 22q11.2, microduplication syndrome 22q11.2 and terminal microduplication syndrome 22q11.2. In this review, we will focus on the 22q11.2 deletion syndrome.

Of all the rare syndromes affecting the same chromosomal area, 22q11.2 deletion syndrome is the most prevalent. The vast majority of studies show that its incidence is one in 4000 live newborns, this number being the same for both sexes [2]. This clinical entity, in addition to being called 22q11.2 deletion syndrome, is also known by several other names, such as “DiGeorge syndrome” and “velocardiofacial syndrome” [1]. Diagnosis for this disease is performed by observing the phenotype and analyzing the clinical signs and symptoms that are present. The definitive diagnosis must be of a genetic nature. 

Ninety percent of cases of DiGeorge syndrome are de novo, although some cases of autosomal dominant inheritance (8–28%) have been reported [2]. 

Variability in gene expression causes differences between patients at the phenotypic level. Craniofacial anomalies and alterations in the development of the thymus and heart stand out [3,4,5].

Therefore, adaptation to daily life of patients with this disease can be significantly affected. The degree of dependence in order to perform daily activities such as talking, due to malformations in the palate, and walking or doing sports normally, due to cardiac manifestations, highlight this aspect. Difficulties in adaptation to the environment and school or work integration are other issues that often have a negative impact on the life of these patients [6].

The main objective of this review is to update the existing knowledge on the 22q11.2 deletion syndrome, producing a solid reference that supports future research on the same topic. This report is important, as it approaches the subject from a broad and generalist perspective, gathering information on a rare disease, about which there are not many studies published in the scientific literature.

## 2. Materials and Methods

### 2.1. Review Protocol

The method used for the preparation of this review was a systematic review of the published scientific literature on 22q11.2 deletion syndrome, which followed the review protocol Preferred Reporting Items for Systematic reviews and Meta-Analyses (PRISMA). This consists of a 27-point checklist on the most representative sections of an original article, as well as the process of developing these guidelines. 

The protocol of this systematic review is available on the PROSPERO website at the link https://www.crd.york.ac.uk/prospero/ (accessed on 28 June 2022) and whose registration number is CRD42021233003.

### 2.2. Eligibility Criteria 

We selected studies published up to July 2022 that provide information on 22q11.2 deletion syndrome, with no restrictions on language of publication or date of publication. There were no restrictions on the type of article, and we accepted all papers published in scientific journals that dealt with the subject of this review. 

### 2.3. Information Sources 

The bibliographic search was carried out in the databases of PubMed, Scopus, OMIM (Online Mendelian Inheritance in Man) [7] and Orphanet [8], a specific database for rare diseases.

The inclusion criteria applied were any type of article or revision, any language, full text and any date. Exclusion criteria regarding date or language were not applied, due to the scarcity of existing scientific publications on the subject. Articles that did not directly address the syndrome were excluded from this study.

### 2.4. Search Strategy

The following table shows the search strings used to perform this review (Table 1).

### 2.5. Data Extraction Process

After performing the search, the articles found were transferred to the Mendeley web application, using the Mendeley web importer tool. They were then organized by folders, according to the database from which they had been obtained, and all duplicates were eliminated. 

Two reviewers independently examined the title, abstract and keywords of each study identified in the search and applied the inclusion and exclusion criteria. For potentially eligible studies, the same procedure was applied to full-text articles. Differences between reviewers were resolved by discussion or by a third reviewer. 

### 2.6. Selection of Studies

After carrying out all the relevant searches in the different databases mentioned, duplicate documents were deleted. Once this was achieved, an initial reading of the title and summary of all the studies found was carried out, to select those that could be related to the information that was intended to be collected. Finally, a detailed reading of each article was carried out, discarding those in which no relationship was found with the objectives and characteristics of this review.

### 2.7. Synthesis of Results

The findings of this review were used to formulate a series of observations that will serve to support subsequent studies, of greater magnitude, on this subject.

## 3. Results

Selection of Studies 

Figure 1 shows a flowchart of how the articles were collected for this literature review.

After an initial analysis according to title and abstract, 119 articles were excluded because there were duplicate articles or articles that did not meet the inclusion criteria. 

Subsequently, after an in-depth reading of the full text of the article, 12 articles were excluded, including systematic reviews that did not provide the necessary information for this review. 

The following table (Table 2) shows a summary of the results of the studies selected for this review.

The studies selected for this review show updated results on the syndrome. They present the extensive clinical signs and symptoms that can occur, such as feeding problems, hypocalcemia and abnormalities in the structure of the heart or palate, which have consequences for patients’ speech ability and their general development [2]. 

Kobrynski and Sullivan [9] report that the specific gene responsible for the clinical presentation of velocardiofacial syndrome, called the *TBX1* gene, was identified in 2001; however, at the locus of chromosome 22 in which the deletion occurs, there are more than 35 other genes. Due to this great genetic variability, individuals affected with DS 22q11.2 present with disorders that affect cognitive, social and emotional aspects, directly influencing their quality of life, compared to that of healthy people and even to individuals suffering from other chronic diseases [3].

Related to the above, Goodwin, McCormack and Campbell [18] mentioned in their study the burden suffered by parents whose children have DS 22q11.2, claiming that they had progressively lost their independence and friendships. The authors included testimonies in which parents confessed that they were socially separated from the rest of the families at school, whose children were healthy. In line with what has been described, Karas et al. [21] mention in their article that families felt they were not prepared to face the care of a person with this disorder, since at certain times they felt stressed and frustrated. In addition, Weisman et al. [19] commented on various aspects of the mother-child relationship in relation to this syndrome. The mothers showed a higher level of intrusiveness, compared to mothers of children belonging to the control group, although the level of affection that the children showed to their mothers was of equal intensity in both groups. On the other hand, it was also observed that the behavioral problems of children with DS 22q11.2 were inversely related to their level of closeness and reciprocity with their mothers.

On the other hand, studies do not only focus on the relationship between parents and their children with the disease. Okashah et al. [20] considered the situation from the perspective and understanding of healthy siblings of infants with DS 22q11.2 syndrome, which they found to be appropriate for their age, considering the complexity of the disorder.

Another factor discussed in one of the collected studies is how socio-economic status affects the cognition and social behavior of children with DS 22q11.2. Allen et al. [22], based on two tests called the *Family Environment Scale–Real Form* (FES-R) and the *Parenting Dimensions Inventory Short Form* (PDI-S) examining general aspects of the family’s social environment and specific parenting approaches respectively, saw that families of children with DS 22q11.2 had great cohesion and organization, and few conflicts arose, although the score on the tests was lower in the domains related to academic level and independence. One of the most striking results was the use of physical punishment with their children as a tool to educate them. This impaired their social skills, academic ease, mental health, and behavior with others. The authors also mention that socio-economic status (SES) can also influence this. The higher the SES, the lower the stress in families and the greater the health in children.

Wagner et al. [10], in their article, also commented on the social functioning of individuals with DS 22q11.2 and how it could be affected, revealing that these children may be exposed to factors that stress them emotionally during their childhood. This could trigger the appearance of anxiety, depression or certain forms of somatization, and lead to a deterioration of their social functioning over time, harming their quality of life.

That same year, Vergaelen et al. [12] published a study that showed an inversely proportional relationship between the level of fatigue in adults with DS 22q11.2 and their quality of life, especially in the psychological aspects and in their environment. In addition, the authors observed that patients’ level of activity decreased and, as a result, so did the quality of their social relationships. Related to this, Looman, Thurmes and O’Conner-Von [13] also investigated cognitive fatigue related to quality of life in children with velocardiofacial syndrome and how it differed from that of healthy children. This was lower in sick infants than in healthy infants and affected males more than females. Other researchers who also showed that quality of life in children and adolescents with DS 22q11.2 is lower than in healthy people and in people with other types of chronic diseases were Joyce et al. [6].

Additionally, other deficiencies have been observed in individuals with DS 22q11.2. Solot et al. [14] refer in their study to the existence of difficulties in communication, which can persist even into adolescence and lead to more advanced problems, such as the deterioration of their social skills and the understanding of concepts and more advanced language. This is due to the deformities of the palate that they experience, which for 67% of children prevents the back of the throat from closing completely when speaking, causing hypernasality and problems when articulating words. Alqarni, Alharbi and Merdad [16], in their clinical case study, suggest that dentists and pediatricians who identify these anomalies should refer them to the appropriate specialists for treatment, which would in turn increase patients’ quality of life.

Written expression can also be impaired. Hamsho et al. [17] investigated possible predictors that could be related to good or poor development of this ability in adolescents with DS 22q11.2. These were working memory and “*set shifting*” (unconsciously shifting attention from one task to another). They added that addressing these two aspects of the executive function of individuals with DS 22q11.2 from childhood with various support programs could be helpful in improving their written expression.

Other aspects of daily life that are affected include schooling and employment. As Mosheva et al. [15] show, the school that is attended (normal or special education) will depend on the cognitive skills individuals possess, so that the education they receive is tailored to their needs. It is stated in the article that the majority of children between 5 and 12 years old attend mainstream schools, and only a minority attend special education schools, while in high school this is reversed.

Finally, and related to everything described so far, Burke and Maramaldi [11] describe in their article that velocardiofacial syndrome could present three criteria that would make it a disorder potentially classified as disabling, according to the Social Security Administration. The criteria comprise the following: that the syndrome is present for at least 12 months, that it results in imminent death or that it has a severe impact on the quality of life of the individual, and that it is also a real impediment to having a good job. The authors concluded that not all individuals suffering from 22q11.2 deletion meet these three criteria and, therefore, they may not be considered disabled, due to the great variability of phenotypes that may present. 

## 4. Discussion

The daily activities of patients with this syndrome are intimately related to the development of their disease, as expressed by Vergaelen et al. in their study of the prevalence of fatigue in patients diagnosed with the syndrome in adulthood [12]. Therefore, delving into the characteristics of the disorder is vital for the correct analysis of aspects related to daily activities. 

Below is a brief description of the disease. 

### 4.1. Epidemiology

22q11.2 deletion syndrome is one of the most common microdeletion chromosomal syndromes in humans, but among the prevalence data described in the literature, there is some controversy due to the underdiagnosis that occurs, as stated by Hacıhamdioğlu et al. in their latest study from 2015 [2]. 

Most studies show that the prevalence of this syndrome is 1 per 4000 live newborns, although other articles indicate that it can vary between 1 per 2000 and 1 per 6395, as confirmed by Kobrynski and Sullivan in their study from 2007 [9]. 

In the same study, it is observed that males and females are equally affected by 22q11.2 syndrome. However, it is more prevalent in some ethnic groups than in others. It has been found to affect Hispanics more than Caucasians, African Americans, and Asians [9]. 

In more than ninety percent of cases of DiGeorge syndrome, deletion occurs randomly during the development of the fetus, although cases of autosomal dominant inheritance are also found. This occurs with a frequency of between 8% and 28%. On the other hand, between 35% and 90% of patients with DiGeorge syndrome and between 80% and 100% of patients diagnosed with velocardiofacial syndrome have the 22q deletion, and this is confirmed by Kobrynski and Sullivan in their study [9]. 

Another factor to be considered is the premature mortality of individuals suffering from this deletion, since approximately 4% of children who suffer from it die between the ages of 3 and 4 months; in adults, it is a cause of premature death at around 40 years old [3]. McDonald et al., in their 2015 systematic review, delve into this aspect and relate it to the quality of life of these patients. 

### 4.2. Aetiology

The cause of this syndrome is most often a 3 million base pair deletion in part of chromosome 22, located on the long arm (q) in the q11.2 region [3]. All humans have two copies of chromosome 22, one inherited from each parent. These is called homologous chromosomes. On each homologous chromosome, there are certain regions called alleles, which are the different versions of the same gene that an individual has. In this syndrome, during ovogenesis or spermatogenesis, a non-allelic meiotic recombination occurs on chromosome 22, recombining the homologous chromosomes 22, but with different alleles from each other, causing the deletion. In most cases, there are no other cases in the family (de novo) but in about 10% of cases the syndrome is inherited from a parent. The risk of recurrence between siblings in a de novo case is 2–3%, due to low-grade parental germline mosaicism. Affected individuals have a 50% risk of having an affected child [23].

In addition, deletions in the DiGeorge region may also occur. Some of these include the TBX1 gene, which plays an important role in the development of the heart, parathyroid glands, thymus and facial structure. The variable expression of the 22q11.2 phenotype is thought to be due to modifier genes on the other allele of 22q11.2 or on other chromosomes [1]. 

### 4.3. Physiopathology

Until recently, the cause of DiGeorge syndrome was not known, but it is now known that it is caused by the deletion of 22q11.2. This has led to the discovery of the main genes involved in the most typical abnormalities that individuals suffering from the disorder usually present with, such as the TBX1 gene. This gene influences the development of the bone structure of the face and neck, the development of large arteries that carry blood outside the heart, the structure of the ears, and also the development of the thymus and parathyroids. The CRKL gene (which is also related to cardiac development), along with others, are also involved [3]. 

However, not all people suffering from the syndrome have the same phenotype and the clinical manifestations are very diverse, due to the variability of expression of deleted genes. 

On the other hand, it has been shown in mice that, in addition to the deletion of the q11.2 locus on chromosome 22, there are also polygenic effects that can affect phenotype variability in patients who have the same deletion on the chromosome. These data presented by McDonald et al. shed more light on variability in humans [3]. 

### 4.4. Clinical Manifestations

Clinical manifestations may vary, depending on the age of the individual. 

Congenital heart defects [5], chronic infections, hypernasal speech, hypocalcemia, feeding difficulties, developmental and speech delays, learning deficits and differences in behavior are typical during childhood. Other less frequent symptoms that may also appear during this period are renal abnormalities, laryngo-tracheo-esophageal abnormalities, hypothyroidism, intrauterine growth restriction, vertebral abnormalities, polydactyly, scoliosis, thrombocytopenia, hearing loss or microcephaly. 

During adolescence and adulthood, on the other hand, behavioral abnormalities can signal that the individual is facing the onset of a mental illness [24,26], which could later lead to the diagnosis of 22q11.2 syndrome. A delay in diagnosis may lead to psychiatric disorders such as anxiety and depression. Hacıhamdioğlu et al., in the conclusions of their study, indicated that psychosocial development of patients affected by this syndrome is notably altered, as is their quality of life related to health [2]. 

### 4.5. Diagnosis

The definitive diagnosis of this syndrome must be justified by genetic study. Primary diagnostic orientation comes from observation and analysis of clinical findings.

Diagnosis of this syndrome can be made by different methods. One of these would be the detection of cardiac anomalies by ultrasound, such as cardiovascular malformations, tetralogy of Fallot, interrupted aortic arch, right aortic arch, truncus arteriosus, ventricular septal defect or patent ductus arteriosus. Vertebral abnormalities may also be detected by X-ray of the cervical spine, checking for scoliosis in the patient, as Burke and Maramaldi report in their article. Another method could be to screen for deletions in the 22q11.2 region in all individuals with palate abnormalities [27]. 

A dysmorphic facial appearance is also a sign of this syndrome, as is the appearance of hypocalcemia during childhood, although these are usually intermittent and usually disappear after the first year of life. Early genetic diagnosis of 22q11.2 deletion syndrome is also very important. This is carried out by performing a FISH or MLPA analysis, to detect microdeletion on the chromosome, once it is suspected that the individual has the syndrome. These analyses would visualize genes or portions of a person’s genes and check for any abnormalities [9]. 

### 4.6. Differential Diagnoses

The differential diagnoses for this disease include the Smith–Lemli–Opitz, CHARGE, Alagille, VATER and Goldenhar syndromes and isotretinoin embryopathy [1], as these are rare diseases that have clinical links with 22q11.2 deletion syndrome. 

### 4.7. Treatment

The treatment implemented during the course of this disorder depends on the anomalies that the patient presents with and that cause symptoms. Several authors deal with this aspect in their studies. Hamosh describes the administration of calcium supplements to combat hypocalcemia, speech therapy to address language deficits [28], psychological therapy if these aspects are affected and, in certain cases, cardiac and palate surgeries may be warranted to solve the anatomical problems that have appeared [27]. There is currently no curative treatment for this disease. The clinical approach focuses on treating the underlying problems that appear with the evolution of the syndrome. The main limitation encountered when studying therapeutic options is the lack of a definitive and effective treatment. Another limitation to highlight is the amount of treatments and their duration since, being a degenerative disease, treatment is required throughout life. The aim is to improve, and sometimes maintain, the highest possible quality of life for these patients [28]. However, the undoubted advances in gene therapy will have a positive impact on the development of this disease, and this may be a line of treatment in future [29].

### 4.8. Prognosis

The prognosis of 22q11.2 deletion syndrome is variable and will depend on the severity of the disorder present. In infant patients, mortality is relatively low (around 4%) and in adults, mortality is higher, compared to the general the adult population [1]. 

### 4.9. Daily Activities

As the results of the articles selected for this review show, the daily activities of patients affected by DS 22q11.2 can be seriously affected, due to the diversity of clinical manifestations and consequences of having this syndrome. There are articles that speak of impairment in aspects of quality of life, such as that of McDonald et al., who describe impairment in cognitive, social and emotional aspects [3].

Socially, patients may experience a decrease in daily activities due to the stress or anxiety they may suffer in childhood, which may carry over into adolescence or adulthood, as they do not have adequate tools to establish stable social relationships. Hacıhamdioğlu et al. commented that this aspect may be aggravated by possible palate anomalies that can prevent sufferers having good verbal communication [2].

Parents and caregivers of children with D 22q11.2 may also be socially isolated, as they are rejected by other families whose children are healthy. This generates in them feelings of discomfort and frustration, so here it is seen that an emotional factor is influenced by the social one. In addition, Goodwin et al. reported in their study that parents and caregivers also tend to lose the friendships they already had, and their independence, by having to take care of their children [18].

As for work, the unemployment rate in adults with intellectual or physical disabilities is higher than in healthy people, so that in many cases they cannot develop independence and economic stability. This is in line with the type of education they have received at school during childhood, since a large percentage do not receive an education adapted to their needs; therefore, they do not develop at the same cognitive level as healthy individuals, and this in turn will also be reflected in their future [15]. Mosheva et al. extended their study from 2019 on education and employment in patients diagnosed with this syndrome [15].

While it is true that the daily activities of these individuals is diminished, this decline can be slowed or improved by taking a series of steps to help solve the problems they face. According to Alqarni et al., in their study on dental management in this type of patient, palate surgeries or going to a dentist during childhood could offer many advantages in terms of verbal communication [16]; in addition, executive function and written expression could be improved with various assistance programs, as indicated in the study by Hamsho et al. [17]. 

Based on the description of the disease [30], an approach that includes both the patient [31] and the family [32] is essential.

## 5. Conclusions

The daily activities of patients with 22q11.2 deletion syndrome are seriously impaired, due to the impact of the clinical manifestations. A series of interventions could be performed to positively influence the improvement of individuals’ social, cognitive and emotional skills, thus increasing their daily capability to perform different activities. Some notable interventions would be to organize tasks in tables to enhance memory, or to incorporate verbal and physical therapies for motor development. In addition to performing a complete analysis of the disorder, we confirm that this syndrome in particular, and rare diseases in general, belong to a clinical area that still has a long way to go and requires the advancement of scientific knowledge to progress. This study proposes a possible line of future research, working on the application of these interventions with patients diagnosed with this pathology, to analyze their impact. This could offer a new perspective on the future of this disease.

## Figures and Tables

**Figure 1 children-09-01168-f001:**
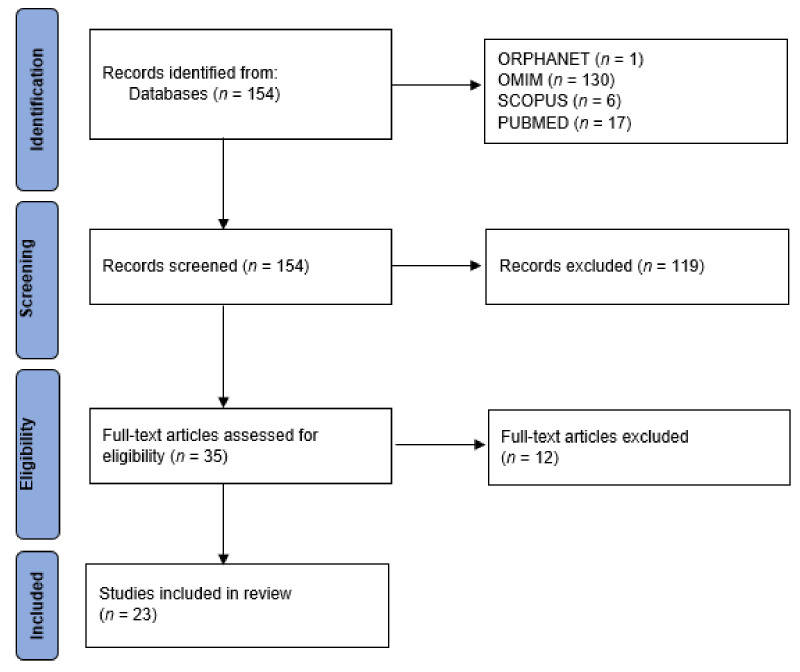
Flowchart showing how articles were collected.

**Table 1 children-09-01168-t001:** Description of search strings.

Sources of Information	Search String
Orphanet	ORPHA:567
OMIM	# 192430 # 188400 # 611867
Scopus	(TITLE-ABS-KEY (22q11.2 AND deletion AND syndrome) OR TITLE-ABS-KEY (digeorge AND syndrome) OR TITLE-ABS-KEY (velocardiofacial AND syndrome) AND TITLE-ABS-KEY (activities AND of AND daily AND living))
PubMed	(((22q11 deletion syndrome[MeSH Terms]) AND (digeorge syndrome[MeSH Terms])) AND (velocardiofacial syndrome[MeSH Terms] AND (activities of daily living[MeSH Terms]). Filters: Full text

The search strings used in the Orphanet and OMIM databases are specific codes for this disease on these platforms. TITLE-ABS-KEY refers to Title-Abstract-Keywords.

**Table 2 children-09-01168-t002:** Summary of the results of the selected studies.

Author	Article	Objectives	Results
Hacıhamdioğlu, Hacıhamdioğlu and Delil (2015) [2]	22q11 deletion syndrome: current perspective	Update the existing knowledge the illness	Updating the technical data sheet on the illness and overview of the syndrome
McDonald-McGinn et al. (2015) [3]	22q11.2 Deletion Syndrome		
Kobrynski and Sullivan (2007) [9]	Velocardiofacial syndrome, DiGeorge syndrome: the chromosome 22q11.2 deletion syndromes		
Wagner et al. (2017) [10]	Childhood predictors of young adult social functioning in DS 22q11.2	Describe results of social functioning, identify predictors in childhood of young adults with DS 22q11.2	Anxiety, depression and somatization during childhood may have a negative impact on social functioning, which is worse in the development periods of these individuals than in healthy siblings and same aged peers
Burke and Maramaldi (2016) [11]	Variability in clinical and anatomical manifestation of velocardiofacial syndrome presents diagnostic and policy uncertainty	Determine if velocardiofacial syndrome fulfilled the 3 criteria of disability for the Compassionate Allowance List (CAL)	The variability of symptoms makes it impossible to claim than all cases of VCFS meet the 3 criteria of disability
Joyce et al. (2018) [6]	Health-related quality of life in 22q11.2 deletion syndrome: the child’s perspective	Explore the quality of children’s and adolescents’ lives with DS 22q11.2 from the child’s perspective and compare that quality of life with that of a healthy group and a group with chronic diseases	Children and adolescents with DS 22q11.2 have a poorer quality of life than healthy children or chronically ill children of the same age
Vergaelen et al. (2017) [12]	High prevalence of fatigue in adults with a 22q11.2 deletion syndrome	Determine the level of fatigue in young adults with DS 22q11.2 in comparison with the normal population. Examine the relationship between the level of fatigue and psychiatric disorders and prevalent somatic disorders in this syndrome and establish if there is any relationship between the level of fatigue and quality of life	Fatigue has been strongly associated with the quality of life scores, especially in the psychological sphere, the environment and in general scores.
Looman, Thurmes and O’Conner-Von (2010) [13]	Quality of life among children with velocardiofacial syndrome	Explore the quality of life of children with velocardiofacial syndrome and compare that quality of life according to gender and with examples of healthy and chronically ill children.	The differences between the quality of life proposed in the study objectives are observed. In addition, the strengths of individuals with DS 22q11.2 are highlighted.
Solot et al. (2001) [14]	Communication in 22q11.2 deletion syndrome: a brief overview of the profile, intervention approaches, and future considerations	Examine potential approaches that may improve the communicative skills of these individuals	It is recognized that the individuals with this syndrome have difficulties with communication, therefore directed interventions must be carried out.
Mosheva et al. (2019) [15]	Education and employment trajectories from childhood to adulthood in individuals with 22q11.2 deletion syndrome.	Explore trajectories of education and employment of individuals with DS 22q11.2, from childhood to adulthood.	Cognitive skills of individuals are more predictive of the type of educational system in which they study than the level of behavioural adaptation, unlike with employment
Alqarni, Alharbi and Merdad (2018) [16]	Dental management of a patient with 22q11.2 deletion syndrome (DS 22q11.2)	Highlight the most common dental characteristics of DS 22q11.2 in order to help pediatric and general dentists with the early detection and treatment of children with DS 22q11.2	There are facial dimorphisms and common features in the dentition, so pediatricians and dentists can use that to refer them to the appropriate specialists in order to increase quality of life
Hamsho et al. (2017) [17]	Childhood predictors of written expression in late adolescents with 22q11.2 deletion syndrome: a longitudinal study	Examine childhood predictive factors of achievement of the written expression in adolescents with SD 22q11.2	Differences were found in the predictive factors between participants of the DS 22q11.2 group and the control group
Goodwin, McCormack and Campbell (2016) [18]	You don’t know until you get there: the positive and negative “lived” experience of parenting an adult child with 22q11.2 deletion syndrome	Describe the perspectives of parents and how they live the experience of having a son with DS 22q11.2.	A variety of feelings and emotions experienced by parents during the upbringing of their children is presented
Weisman et al. (2015) [19]	Mother-child interaction as a window to a unique social phenotype in 22q11.2 deletion syndrome and in Williams syndrome	See the differences between mother-child relationships in individuals with DS 22q11.2, Williams Syndrome and developmental disability (disorders of the development).	The differences in mother-child relationships in these syndromes, compared to each other
Okashah et al. (2014) [20]	Parental communication and experiences and knowledge of adolescent siblings of children with 22q11.2 deletion syndrome	Explore what understanding healthy siblings of individuals with DS 22q11.2 have of this disease. Determine the frequency, method, and content of information that parents give to their unaffected children.	In general, parents feel that their children that have not been affected by this syndrome have appropriate knowledge for their age
Karas et al. (2014) [21]	Perceived burden and neuropsychiatric morbidities in adults with 22q11.2 deletion syndrome.	Know the experience and needs of caregivers of this type of patients, especially during the period of transition from childhood to adulthood.	The burden they suffer for being caregivers of this type of patients is great. In addition, a degree of dissatisfaction with certain services provided to these individuals is common
Allen et al. (2014) [22]	Association of the family environment with behavioural and cognitive outcomes in children with chromosome 22q11.2 deletion syndrome	Examine the impact of family environment on social behaviours and cognitive results of the pediatric DS 22q11.2 population.	It determined that the socio-economic status, parental control and family organization affect the social behavior and cognitive outcomes of children with DS 22q11.2
Funato, N.(2022) [4]	Craniofacial Phenotypes and Genetics of DiGeorge Syndrome	Update existing information on the disease through a systematic review.	Comprehensive study on the phenotype and genotype in humans and mouse models for this disease
Zhang et al. (2021) [23]	Chromatin Modifications in 22q11.2 Deletion Syndrome	Identify the duration of chromatic change and its influences on cellular behavior in this pathology.	B cells had a minimally altered epigenetic landscape in 22q11.2
Butensky et al. (2020) [5]	Cardiac evaluation of patients with 22q11.2 duplication syndrome	Study the cardiac evolution in patients diagnosed with the syndrome	Minor cardiac malformations were infrequently identified in patients without previously known congenital heart disease
Buijs et al. (2020) [24]	Cognitive behavioral therapy in 22q11.2 deletion syndrome: A case study of two young adults with an anxiety disorder	Describes case report on the pathology	Two diagnosed cases, under treatment and phenotypic description
Correa et al. (2020) [25]	Identification of relevant International Classification of Functioning Disability and Health (ICF) categories in patients with 22q11.2 Deletion Syndrome: a Delphi exercise.	Identify the category with the greatest relevance to this disease in the International Classification of Functioning Disability and Health (ICF)	A list of the different categories analyzed
Santambrogio et al. [26] (2022)	Psychiatric and psychological support for an adolescent woman with chromosome 22 deletion syndrome and intellectual disability: a good outcome	Describes case report on the pathology	Case report of a woman with chromosome 22 deletion syndrome (22q11.2DS), mild intellectual disability (ID) and associated psychiatric disorders

## Data Availability

Not applicable.

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
