# Peer review of "Deletion Syndrome 22q11.2: A Systematic Review"

_children, 2022, doi:10.3390/children9081168_

Round 1
Reviewer 1 Report
I have to reject this review.
I'll write down some comments/suggestions since I think that with a little bit of effort, this manuscript could be re-submitted and maybe published since you are right about the fact that not all the available literature about the topic are really consistent and informative.
First of all, since it's a review you are going to publish in 2022, in my opinion, it is not acceptable to update the state of art about the topic up to Dicember 2020. If you search for other publications during 2021 and 2022, you will see that there has been a good and useful update about this rare disease.
Beeing a review, there are very few references.
For most of the body of the manuscript, there are informations not updated with a caotic organization.
I find that in the abstract there are too many rows about materials and methods that you repeat in the text of concerning section.
For "Materials and Method"section, I find opinable almost everything you wrote. About the tools or databases you have used you could summarize with graphs or figures and put in the text only the settings you used (if necessary and not present in the captions), the tools/database you used, the version, of course the references. You may also use other kind of databases to collect other informations to turn the manuscript more informative.
Author Response
First of all, since it's a review you are going to publish in 2022, in my opinion, it is not acceptable to update the state of art about the topic up to Dicember 2020. If you search for other publications during 2021 and 2022, you will see that there has been a good and useful update about this rare disease.
We updated the literature up to the requested date.
Beeing a review, there are very few references.
Increased the number of references
For most of the body of the manuscript, there are informations not updated with a caotic organization.
We improved the organization of the manuscript.
I find that in the abstract there are too many rows about materials and methods that you repeat in the text of concerning section.
We improve the abstract as requested.
For "Materials and Method"section, I find opinable almost everything you wrote. About the tools or databases you have used you could summarize with graphs or figures and put in the text only the settings you used (if necessary and not present in the captions), the tools/database you used, the version, of course the references. You may also use other kind of databases to collect other informations to turn the manuscript more informative.
We improve this section.
Reviewer 2 Report
The systemic review article by Jonathan et al have updated the review of literature on 22q11.2 deletion syndrome. The reported review is helpful to understand the etiology and pathophysiology of the disease. However, the authors need to address the following concerns
Concerns
1. The major caveat of this review is limited information, would be better to update the literature to date
2. Need to expand the underlying molecular pathogenesis and limitations of the therapeutic options
Author Response
The systemic review article by Jonathan et al have updated the review of literature on 22q11.2 deletion syndrome. The reported review is helpful to understand the etiology and pathophysiology of the disease. However, the authors need to address the following concerns
Concerns
- The major caveat of this review is limited information, would be better to update the literature to date
We updated the literature up to the requested date.
- Need to expand the underlying molecular pathogenesis and limitations of the therapeutic options
We provide more information on both aspects requested.
Round 2
Reviewer 1 Report
Since last version I rejected, you clearly implemented your manuscript.
However there are some things that in my opinion could be improved.
rows 79-82
I think it is not essencial to specify the boolean operators you used or the search strings since they are almost obvious and you repeat it in the Table 1 in "search strategy". At least conserve just one of the two parts in which you explain your search strategy.
In materials and methods you should specify the reference and the version of the tools or databases you used.
e.g. OMIM: what is? What does the acronymous mean? Which is the version (I read that this version is Updated July 30, 2022)
Does it have a citation?
page 4 formatting problems
Improve the Discussion and or Conclusion with some future perspectives.
Author Response
Since last version I rejected, you clearly implemented your manuscript.
However there are some things that in my opinion could be improved.
rows 79-82
I think it is not essencial to specify the boolean operators you used or the search strings since they are almost obvious and you repeat it in the Table 1 in "search strategy". At least conserve just one of the two parts in which you explain your search strategy.
We eliminate duplicate information
In materials and methods you should specify the reference and the version of the tools or databases you used.
e.g. OMIM: what is? What does the acronymous mean? Which is the version (I read that this version is Updated July 30, 2022)
Does it have a citation?
We add requested information
page 4 formatting problems
We corrected formatting errors
Improve the Discussion and or Conclusion with some future perspectives.
We add future perspectives in both sections
Reviewer 2 Report
Thank you for answering the raised concerns and I don't have any further questions.
Author Response
Thank you for answering the raised concerns and I don't have any further questions.
Thank you for your time and attention.